# A Crosstalk between the Cannabinoid Receptors and Nociceptin Receptors in Colitis—Clinical Implications

**DOI:** 10.3390/jcm11226675

**Published:** 2022-11-10

**Authors:** Maria Wołyniak, Ewa Małecka-Wojciesko, Marta Zielińska, Adam Fabisiak

**Affiliations:** 1Department of Digestive Tract Diseases, Faculty of Medicine, Medical University of Lodz, 90-153 Lodz, Poland; 2Department of Biochemistry, Faculty of Medicine, Medical University of Lodz, 92-215 Lodz, Poland

**Keywords:** nociceptin opioid receptor, endocannabinoid, colitis, inflammatory bowel diseases, inflammation, nociception

## Abstract

Inflammatory bowel diseases (IBD) refer to a group of gastrointestinal (GI) disorders with complex pathogenesis characterized by chronic intestinal inflammation with a variety of symptoms. Cannabinoid and nociceptin opioid receptors (NOPs) and their ligands are widely distributed in the GI tract. The nociceptin opioid receptor is a newly discovered member of the opioid receptor family with unique characteristics. Both cannabinoid and NOP systems exhibit antinociceptive and anti-inflammatory activity and contribute to maintaining proper motility, secretion and absorption in the GI tract. Furthermore, they influence high and low voltage calcium channels, which play a crucial role in the processing of pain, and share at least two kinases mediating their action. Among them there is NF-κB, a key factor in the regulation of inflammatory processes. Therefore, based on functional similarities between cannabinoid and nociceptin receptors and the anti-inflammatory effects exerted by their ligands, there is a high likelihood that there is an interaction between cannabinoid receptors 1 and 2 and the nociceptin receptor in colitis. In this review, we discuss potential overlaps between these two systems on a molecular and functional level in intestinal inflammation to create the basis for novel treatments of IBD.

## 1. Introduction

The newly discovered nociceptin opioid receptor (NOP) and its endogenous ligand nociceptin are habitually classified as a ‘non-classical’ part of the opioid family due to the fact that their molecular mechanisms differ from other members of this group [1]. In addition to interacting with cannabinoid receptors 1 and 2 (CB1 and CB2, respectively), the endogenous cannabinoid system (ECS) communicates with other receptor families such as opioid receptors [2]. Moreover, the co-administration of cannabinoids and opioids was proven to produce synergistic effects in pain management [3]. Little is known on the potential crosstalk between CB receptors and NOP receptors. There is a high likelihood that the exploration of this phenomenon could shed new light on the future treatment of diverse diseases.

Inflammatory bowel diseases (IBD) are a group of the gastrointestinal (GI) disorders with complex pathogenesis comprised mainly of Crohn’s disease (CD) and ulcerative colitis (UC). Genetic, environmental, immunological and infectious factors contribute to the development of IBD [4]. Although the reduction in colectomy rates in recent decades was achieved due to better care, the symptoms of IBD often persist, and there remains a pressing need to improve the quality of patients’ lives. Thus, there is ongoing research on IBD pathogenesis, and numerous drugs are under investigation. The potential antinociceptive and anti-inflammatory activity of NOP agonists may fill this gap and appear to be a reasonable treatment option for patients with GI disorders. In addition, with the evident crosstalk between cannabinoid and opioid receptors, NOPs may fit this pattern, but this has not yet been fully explored. Further investigation on shared patterns could establish a basis for the enrichment of IBD therapy. In this review, we elaborate on the possible interactions between CB receptors and NOP receptors during intestinal inflammation.

## 2. Endogenous Cannabinoid System

The ECS consists of classical receptors, CB1 and CB2, and their endogenous ligands, endocannabinoids. There are also various compounds involved in the modulation, synthesis and degradation of the latter. Two best described endocannabinoids, N-arachidonoyl-ethanolamine (AEA; anandamide) and 2-arachidonoylglycerol (2-AG) are non-selective agonists, with a higher affinity with CB1 than with CB2 [5,6]. The group of classical cannabinoids can be divided into compounds that occur naturally in *Cannabis Sativa* or synthetic analogues of these compounds. Δ9-THC and Δ8-THC are psychotropic constituents of cannabis, while the most investigated synthetic cannabinoids are 11-hydroxy-Δ8-THC-dimethylheptyl (HU-210) and desacetyl-L-nantradol [7]. Aminoalkylindole compounds typified by R-(+)-WIN55212 are commonly used agonists of CB [8].

CB1 and CB2 are two classical cannabinoid receptor types. Both receptors are coupled to pertussis toxin-sensitive G protein to inhibit adenylyl cyclase activity and to initiate the molecular pathway combined with mitogen-activated protein kinase (MAPK) [6]. These two receptors differ in terms of amino-acid sequence, signaling mechanisms and tissue distribution. CB1 is mostly present in the central nervous system (CNS), particularly in the cortex, basal ganglia, hippocampus and cerebellum [9]. The CB2 receptor is expressed at a lower rate in the CNS compared to CB1 and is peripherally abundant, mostly in the vascular system, immune cells and in the enteric nervous system (ENS) [9]. Recent findings suggest that cannabinoids can also activate other receptors, including GPR55, GPR18, GPR 119 and the vanilloid receptor TRPV1 [10,11,12,13]. The receptors listed above, referred as non-classical cannabinoid receptors, share common molecular pathways, ligands and tissue distribution with CB1 and CB2 and play a role in the same pathological conditions [14].

In the United States, drugs that activate CB1and CB2 receptors are already commercially available: Cesamet (nabilone), Marinol (dronabinol, D9-tetrahydrocannabinol (D9-THC)) and Sativex (D9-THC with cannabidiol). The first two can be prescribed to reduce chemotherapy-induced nausea and vomiting. Marinol can be prescribed to stimulate appetite, while Sativex is prescribed as an analgesic for cancer pain and the symptomatic relief of neuropathic pain in adults with multiple sclerosis [15,16]. However, data on the efficacy of treatment using cannabis are deficient and inconsistent, and the therapy itself remains controversial. Currently, there is a lack of well-established and approved therapy based on cannabinoids in IBD, but their medical usage has increased due to tendency among patients towards complementary and alternative medicine (CAM) [17]. Research based on surveys collected among Spanish patients reported 40–50% rates of CAM usage among IBD patients [18]. Ten percent of them admitted to taking mainly cannabis derivatives. Interestingly, patients diagnosed with UC tended to have a higher rate of CAM usage than CD patients [18].

Endocannabinoid receptors and their respective ligands are widely distributed in the GI tract and mediate inhibitory pathways by reducing the activity of the vagal nerve. CB1 receptor activation is associated with an improvement in appetite, relief from nausea and vomiting and the deceleration of intestinal motility, while the role of CB2 is to modulate inflammation [8]. The majority of those observations, in addition to evidence that the modulation of CB receptors is efficient in the alleviation of intestinal inflammation, have been demonstrated in animal models of colitis [8]. In a study performed by Massa et al. [19], the CB1 agonist HU210, at the dose of (0.05 mg/kg) injected 30 min before and 24 h and 48 h after 2,4-dinitrobenzene sulfonic acid (DNBS) infusion, protected against experimental colitis, while the CB1 antagonist SR141716A (3 mg/kg) enhanced the extent of intestinal inflammation [19]. Additionally, Singh et al. [20] showed that the CB2 agonist JWH-133 effectively attenuated inflammation in the mouse model of colitis induced by dextran sodium sulphate (DSS) and in interleukin (IL)-10 deficient mice when compared to the control. However, the use of CB1 and CB2 ligands in clinical practice is limited due to undesired side effects affecting the CNS, such as hypothermia, catalepsy, the inhibition of activity or impaired ambulation. The crosstalk between ECS and opioid receptors, specifically mu, delta and kappa (MOP, DOP and KOP, respectively), was extensively studied [2]. This activity occurs on different levels ranging from direct receptor–receptor interactions to similarities in signal transduction pathways [21]. Salaga et al. [22] showed that PR-38, a salvinorin A analogue, alleviated intestinal inflammation by acting via cannabinoid and opioid receptors, with no binding affinity to CB sites in vitro [22].

## 3. NOP Receptor

The NOP receptor was discovered in 1994 in mice as an orphan receptor. Its endogenous ligand heptadecapeptide nociceptin/orphanin FQ was simultaneously described a year later by two independent groups and named nociceptin and orphanin, respectively [23,24]. Its transcripts are abundant in the CNS, mostly in the forebrain, brain steam and in both the dorsal and ventral horns of the spinal cord, with NOP receptors also being distributed in the smooth muscles, peripheral ganglia and the immune system [25], suggesting a wide range of functions. NOP receptors became a point of interest of various research groups and were claimed to play a role in food intake by the elevation of food consumption (when administered centrally) [26], prolonged stress and post-traumatic stress disorder [27], learning [28] and addiction, including alcohol abuse [29]. NOP receptor agonists are used in preclinical models of anxiety, cough, substance abuse, pain and the micturition reflex. The administration of nociceptin into the lateral ventricle or the bed nucleus of the stria terminalis prevents the anxiogenic effect of intracerebroventricular (i.c.v) corticotropine-releasing factor injections [30]. NOP receptor antagonists could constitute the foundation for the treatment of analgesia, depression and motor symptoms in Parkinson’s disease [31,32].

Similar to CB receptors and other opioid receptors, NOP receptors were associated with the group of G protein-coupled receptors (GPCRs), which regulate adenylate cyclase activity. NOP receptors exhibit over 60% similarity in terms of sequence with other opioid receptors [33]. Correspondingly, the NOP receptor ligand, nociception, resembles the structure of other opioid ligands (especially dynorphin A) [34] and it does not bind to classical opioid receptors [35]. The analysis of 3.0A resolution X-ray structures revealed the key distinction between NOP receptors and opioid receptors [33]. The amino-terminal tetrapeptide Phe-Gly-Gly-Phe in NOP receptors is similar to the Tyr-Gly-Gly-Phe sequence found in classical receptors. The aromatic side chain in a first position (Phe) does not interfere with the action of NOP receptors, whereas in other opioid receptors, any change in the first Tyr position leads to complete loss of activity [32]. In addition, NOP receptors appear to require complete peptide molecule recognition in order to be activated [33].

The sustained study of the NOP receptor and its endogenous ligand arises from continuous demand for compounds that could target opioid receptors without the undesired side effects caused by opioids. Among them, the most significant is the withdrawal effect, which decrease after the activation of NOP receptors in the CNS. To evaluate this effect, a group of rats underwent repeated daily intragastric ethanol administrations to become substance-dependent. Later, they were divided into three groups and received nociceptin at doses of 0.01, 1.0 and 3.0 μg/rat i.c.v. 10 min before behavioral measures of ethanol withdrawal [36]. The results showed that the activation of NOP receptors in the CNS significantly attenuated the expression of somatic withdrawal signs in ethanol-intoxicated rats. For most of the somatic withdrawal signs, the abstinence scores significantly reduced during the 12 h observation period [36]. Based on previously described examples, the unique traits of the NOP system indicate their possible use in the treatment of many different pathologies.

## 4. Possible Crosstalk between ECS and NOP

Assuming the functional similarities between CB and NOP receptors and their ligands, summarized briefly in Figure 1, the hypothesis of a crosstalk between those two systems has emerged [35]. The major mediators of CB receptors are the G proteins of the Gi/o family. These inhibit adenylyl cyclase activity in most tissues and cells and regulate ion channels, including K^+^ and Ca^2+^ ion channels, which serves as an important component of neurotransmission modulation [37]. However, recent studies revealed that CB1 can also stimulate adenylyl cyclase via Gs, induce receptor-mediated Ca^2+^ fluxes and stimulate phospholipases C and A in several experimental models [38,39]. Derkinderen et al. [40] showed that lysophosphatidic acid increased the phosphorylation of p38 MAPK and c-Jun N-terminal kinase in rat and mouse hippocampal slices, and pre-treatment with a selective antagonist of CB1, SR 141,716, at a concentration of 100 µM per slice, abolished these effects [40]. The stimulation of CB1 and CB2 receptors leads to the phosphorylation and activation of p42/p44 mitogen-activated protein kinase MAPK (ERK), p38 MAPK and Jun N-terminal kinase (JNK) as signaling pathways to regulate transcription in the nucleus, as described broadly in a review by Kendall et al. [41]. Moreover, the action of CB receptors is associated with β-arrestin molecules, which cause the formation of signaling complexes which regulate GPCR signaling [6].

With regard to the NOP receptor and its endogenous ligand nociceptin, our knowledge of the direct molecular mechanisms involved in the regulation of their action is constantly expanding. The NOP receptor is a Gi/o protein-coupled receptor that inhibits adenylate cyclase and regulates K^+^ and Ca^2+^ channels, inhibiting neurotransmission [1]. The involvement of ERK, JNK, p38 and NF-κB in the regulation of prepronociceptin and NOP receptor signaling pathways was tested with specific kinase inhibitors in human peripheral leukocytes in inflammatory conditions. A selective blockage of ERK and p38 MAPK with selective inhibitors (PD98509 (30 μM) and SP600125 (10 μM), respectively) completely prevented the phorbol-12-myristate-13-acetate-induced downregulation of mRNA NOP expression. Therefore, the two kinases which play a key role in the regulation of nociceptin and its receptor signaling are ERK and p38, which correspond to the mechanisms of cannabinoid pathways described above [42,43].

Concurrently, nociceptin has been shown to increase the phosphorylation of IκB kinase, promoting the phosphorylation and degradation of IκB in SH-SY5Y human neuroblastoma cells, thus suggesting that nociceptin may modulate signaling when combined with nuclear factor kappa B (NF-κB) [44]. Recently, Donica et al. [45] found that nociceptin modulates NF-κB activity on SH-SY5Y cells, as the nuclear accumulation and the DNA binding of NF-κB is both protein kinase C-dependent and NOP receptor-specific. Moreover, NF-κB inhibitors interfered with the effects of the lipopolysaccharide-induced expression of nociceptin, and subunits p50 and p65 of NF-κB were shown to form a heterodimer which is implicated in inflammatory pathways [45,46]. It was also demonstrated that the stimulation of NOP receptors leads to c-Jun N-terminal kinase activation, the phosphorylation of c-Jun, the activation of transcription factor-2 [47] and the stimulation of phospholipase A [48] and C [49]. Finally, β-arrestin was also demonstrated to be crucial in NOP signaling [45]. The above shared molecular pathways are presented in Figure 2.

CB and NOP receptors, when activated by their ligands, influence the function of voltage-gated ion channels. Inwardly rectifying potassium channels, important in controlling membrane potential, are either stimulated (tested in vitro on HEK293 cell line and mouse aortic myocytes) or inhibited (tested in vitro on rat ventricular myocytes) by anandamide [50]. Nociceptin was found to induce a concentration-dependent reversible outward current in melanin-concentrating hormone neurons. The effect persisted in tetrodotoxine, was reversed near the potassium equilibrium potential and displayed inward rectification, suggesting direct postsynaptic potassium channel activation [51].

High voltage activated L-type calcium channels are expressed in the cardiovascular system, when they are critical for the formation of cardiac action potential. They also can be found on neuronal cell bodies, where they mediate calcium-dependent gene transcription and control, an action of calcium-dependent enzymes such as ghrelin, a gastric peptide hormone which itself increases voltage-activated calcium currents [52,53]. L-type calcium channel blockage by the CB1 receptor agonist WIN55122 was described in neurons of neonatal rat solitary tracts [54], while its activation was achieved using the CB receptor agonist DALN in N18TG2 neuroblastoma cells [55]. In terms of nociceptin, in a study concerning an effect of postsynaptic opioid and cannabinoid receptors on calcium currents in a neonatal rat nucleus of the solitary tract, it was concluded that N and P/Q type calcium channels were inhibited by both DAMGO and nociceptin, while L-type channels were inhibited by WIN55122.7. Those observations suggest that MOP, KOP and NOP receptors inhibit N- and P/Q-type channels by Gα-protein βγ subunits, whereas CB1 receptors inhibit L-type channels mediating through Gα-proteins involving protein kinase A [54].

T-type calcium channels are distributed in the central and peripheral nervous system. These channels control neuronal excitability, playing a crucial role in neural activities and are linked to chronic neuronal disorders such as epilepsy, hypertension and pain [56]. Anandamide, a CB receptor agonist, tends to inhibit T-currents, even independently from the activation of CB1/CB2 receptors, G-proteins, phospholipases and protein kinase pathways. It stabilizes T type calcium channels in the inactive state and is responsible for a significant decrease in calcium currents associated with neuronal firing activities [53]. Similar to the case of cannabinoid ligands, in a study of medium-sized L4–L5 dorsal root ganglion neurons from rats, it was found that nociceptin administered at a concentration of 0.01–1.0 mM suppressed low-voltage-activated transient calcium currents, which confirms that nociceptin can diminish neuronal excitability [57]. Additionally, nociceptin was found to modulate cell signaling in the G-protein-independent mechanism, in contrast to morphine. Although both agonists suppressed high-voltage-activated Ca^2+^ channel conductance via a G protein-dependent mechanism, only nociceptin selectively affected the low voltage-activated transient calcium channel, and this may be crucial to understanding why its behavioral actions differ from those of morphine and other common opioid receptor agonists [57].

N-type calcium channel influx plays a key role in the regulation of nociceptive signaling in the spinal cord. Pain-sensing dorsal root ganglion neurons and their synaptic connection with neurons of the spinothalamic tract rely almost completely on N-type channels; thus, the inhibition of N-type channels in these neurons via the activation of opioid receptors consequently produces analgesia [53]. With reference to cannabinoids, it was established that N type voltage-gated calcium currents in the neuroblastoma-glioma cell line NG108-15 are inhibited by WIN55212-2, which decreases excitability and neurotransmitter release [58]. Correspondingly, the NOP receptor was found to inhibit N-type calcium channels, even without binding nociceptin [59]. Additionally, the prolonged exposure of NOP receptors to their endogenous ligand expressed in Chinese hamster ovary cells triggers calcium channel internalization into vesicular compartments in a protein kinase C-dependent manner. This phenomenon was described selectively between N-type calcium channels and NOP receptors [59]. However, the results of a recent study, in which the authors tested this mechanism on rat dorsal root ganglia using whole-cell patch-clamp recordings of N-type calcium channels, showed that nociception-prolonged action on NOP receptors does not cause the internalization of N-type calcium channels [60]. Based on inconclusive results, this potentially promising of modulating pain needs to be explored.

The potential synergistic effects of cannabinoids and classical opioids was tested repeatedly and the results stated that the administration of various cannabinoids with morphine produces a synergistic effect as measured when applying the tail flick test in mice [61] and in rhesus monkeys when measured via warm water test withdrawal [62]. The effect of the co-administration of cannabinoids and morphine on nociceptive behavior in a rat model of persistent pain was studied in the thalamus, and the intraparietal pre-administration of Δ-9-THC (1 or 2.5 mg/kg), cannabidiol (5 mg/kg), morphine (2 mg/kg), Δ-9-THC + morphine, Δ-9-THC + cannabidiol or vehicle was evaluated in the formalin-evoked nociceptive behavior test. The results showed that, when administrated selectively, both Δ-9-THC and morphine reduced both phases of formalin-evoked nociceptive behavior, while cannabidiol alone had no effect. The intraparietal co-administration of Δ 9-THC and morphine reduced the second phase of formalin-evoked nociceptive behavior in rats more significantly than either drug alone [63]. However, despite a significant synergistic effect, data concerning cannabinoids (administrated alone or combined with opioids) as a potential solution to alleviate opioid withdrawal syndrome or the development of tolerance are contradictory and inconsistent [62,64,65]. As was mentioned above, an action of both CB and NOP receptors is associated with pain relief, sedation, memory, cognitive functions, locomotor activity and thermoregulation [61,66]. Therefore, potential crosstalk between two systems could create a basis for the enhancement of either NOP receptors/nociceptin or ECS effects in various areas.

In a study by Cichewicz et al. [67] regarding the correlation between the formation of morphine tolerance and the protein expression of the CB1 receptor, no apparent changes in the amount of CB1 were described in midbrain regions, and there was an up-regulation of CB1 receptors in the spinal cord [67]. Contrarily, Cannarsa et al. [68] showed that exposure to Δ9-THC at doses of 100, 150 and 200 nM for 24 h reduced the expression of NOP receptors in neuroblastoma SHSY5Y cells. Moreover, Δ9-THC caused a dose-dependent decrease in NOP receptor mRNA levels. Both effects were blocked by a selective CB1 antagonist, AM251, demonstrating that Δ9-THC is able to affect NOP receptors and that the mechanism is most likely activated by the stimulation of the CB1 receptor. Additionally, an effect triggered by ECS on the nociceptin system varies from the one exerted on traditional opioids [68].

It is well established that classical opioid receptor activation causes primary hyperthermia and later hypothermia in a dose-dependent manner. With regard to the NOP receptor system, nociceptin administered once solely into the brain at a dose of 9–18 µg induced hypothermia in adult rats. Moreover, nociceptin injected at a dose of 1.8 µg 30 s after the injection of morphine at a dose of 4 mg/kg decreased morphine-induced hyperthermia [69]. To investigate the interactions between ECS and NOP receptors in the context of hypothermia, cannabinoid agonist WIN 55212-2 (at a dose of 2.5, 5 and 10 mg/kg, intraperitoneally (i.p.)), selective cannabinoid CB1 antagonist SR 141716A (at a dose of 5 mg/kg, intramuscularly (i.m.)) and NOP receptor antagonist JTC-801 (1 mg/kg, i.p.) were administered solely or simultaneously in various configurations to conscious rats. The authors established that JTC-801 diminished a significant proportion of hypothermia caused by non-selective CB receptor activation. Based on these findings, it is suggested that NOP receptor activation is needed for cannabinoids to produce a full hypothermic response [70]. Interestingly, the hypothermic effect of nociceptin was blocked by JTC-801 but not by selective SR 141716A [70].

Revealing the molecular mechanisms of the crosstalk between ECS and NOP receptors has the potential to change approaches to pain management. To describe the connection between the CB and NOP receptors, research was carried by Gunduz et al. [71], who evaluated the anti-allodynic effect of both the cannabinoid agonist WIN 55212-2 (0.1–10 mg/kg, i.p.) and the NOP antagonist JTC-801 (0.1–10 mg/kg, i.p.). The results revealed that the co-administration of WIN 55212-2 and JTC-801 in mice with neuropathic pain caused by partial sciatic nerve ligation produced greater anti-allodynic effects compared with each compound administered alone. It has been suggested that the increased activity of the NOP system could be the reason for the lower responsiveness of opioids in neuropathic pain. In addition, the authors suggested that NOP receptor antagonists could increase the analgesic effect of morphine in neuropathic pain [71]. Additionally, the chronic administration of JTC-801, at a dose of 1 mg/kg, i.p, attenuated the development of tolerance to the antinociceptive effect of WIN 55212-2 at a dose of 4 mg/kg, i.p. in rats. Moreover, nociceptin levels significantly increased in the amygdala, periaqueductal gray, nucleus raphe magnus and locus coeruleus of rat brains when WIN 55212-2 was combined with JTC-801. Based on these findings, it was hypothesized that NOP receptor antagonism prevents the development of tolerance to cannabinoid antinociception [72].

## 5. Possible Use of NOP Receptors in Intestinal Inflammation

Abdominal pain, including visceral pain, is present in up to 70% of IBD patients in both flare and remission stages [73]. The nociceptin system is abundantly present in the human GI tract [25,74], and NOP receptors, like other opioid receptors, play a role in the maintenance of GI homeostasis, affecting the secretion of hormones and neurotransmitters and decreasing intestinal motility [75]. There is a high possibility that NOP receptors and nociceptin can be used in the treatment of disorders associated with accelerated GI transit and visceral pain such as IBD.

Various therapies including drugs (antispasmodics drugs, nonsteroidal anti-inflammatory agents (NSAIDs) and opioids) and psychological and physical methods (relaxation, acupuncture, hypnosis and the use of heat and cold) are utilized by patients suffering from IBD [76]. The most commonly used drugs for pain management are nonselective NSAIDs or more selective COX-2-inhibiting drugs (i.e., coxibs) and narcotic analgesics. NSAIDs or COX-2 inhibitors are used in IBD patients mostly due to extraintestinal manifestations of IBD: ankylosing spondylitis, arthralgias or peripheral arthritis and osteoporosis with an increased risk of fractures. However, there are rising concerns associated with using NSAIDs because of growing evidence of a risk of deterioration and relapse of IBD with the long-term usage of these drugs [77]. The effect is mostly associated with a reduction in prostaglandin concentration levels, but the data remain conflicting [77].

As a consequence, 21% of outpatients and 62% of hospitalized patients suffering from IBD are chronically treated with narcotics [78]. The problem is that long-term opioid use is connected to a wide range of side effects, with the most harmful among them being drug abuse and dependence. In addition, many other side effects have been reported, such as nausea, reduced GI motility and narcotic bowel syndrome, which refers to chronic abdominal pain aggravated by narcotic use. Also, the data from the prospective Crohn’s Therapy, Resource, Evaluation, and Assessment Tool registry of more than 6000 patients suggest that the use of narcotic analgesics is associated with higher overall mortality [76].

Little is known about the precise molecular mechanisms of pain in IBD, but few have been described. The intestinal innervation includes both intrinsic and extrinsic components of the autonomic nervous system. Intrinsic innervation is controlled by the enteric nervous system, which is a complex connection of nerve plexuses between the muscular and submucosa layers of the intestinal wall. The enteric nervous system controls the secretion of acetylcholine and neuropeptides, absorption and secretion and GI motility, which, when disturbed, contributes to IBD symptoms such as constipation or diarrhea but not to pain transmission [79]. Signals, which are consciously interpreted as pain, are conveyed from the lower GI tract to the CNS by two major sensory afferent nerve trunks, the splanchnic and pelvic nerves [79]. GI inflammation modulates neuronal properties in nociceptive dorsal root ganglia, which innervate the GI tract via the exhibition of hyperexcitability characterized by a decreased threshold for activation and changes in voltage-gated Na^+^ and K^+^ channels [80].

The role of NOP receptors in the inhibition of acetylcholine expression, interference with voltage-gated K^+^ and Ca^2+^ ion channels and interaction with opioid and cannabinoid systems was described above. To test how this interplay flows in a mouse model of pain induced by chronic constriction injury to the sciatic nerve, Zajaczkowska et al. [81] used metamizole, one of the non-opioid analgesics commonly applied in clinical practice in the treatment of somatic and visceral pain, and buprenorphine, a partial agonist of MOP receptors, to observe its effect on various nociception receptors. It was established that metamizole at a dose of 500 mg/kg i.p. decreased spinal levels of NOP but did not alter the expression of other members of the opioid receptor family, MOP, DOP and KOP, or other important receptors involved in nociception: transient receptor potential vanilloid 1 (TRPV1) and transient receptor potential ankyrin 1 (TRPA1). It was suggested that buprenorphine may reduce the analgesic effect of metamizole through NOP receptors [81].

There were also attempts to establish new IBD therapies targeted at the particular accompanying pathologies and symptoms. The study involved in vitro and in vivo experiments as well as the quantification of endogenous nociceptin levels in the serum of IBD patients. To test the effect of the activation of NOP receptors on smooth muscle contractility in the mouse ileum and colon, organ bath studies on isolated tissue sections were performed [82]. The NOP selective ligand SCH 221510 and selective agonist NOC(1–13)NH2, both at a concentration of 10^−10^–10^−6^ M, significantly inhibited EFS (8 Hz)-induced contractions in a concentration-dependent manner. Using intestinal transit and colon bead expulsion tests, it was shown that SCH 221510 at a dose of 1 mg/kg, i.p. significantly increased the colon bead expulsion time and the upper GI transit time [82]. Additionally, SCH 221510 at doses ranging from 0.03 to 1 mg/kg, i.p. dose-dependently inhibited GI motor hyperactivity in mouse models of hypermotility and diarrhea. Moreover, SCH 221510 administered at a dose of 1 mg/kg, i.p. significantly decreased pain-induced behaviors in a mustard oil (MO)-induced mouse model of pain [82]. Finally, based on tissues collected from IBD patients, it was demonstrated that NOP mRNA expression in colon biopsies and nociceptin levels in serum from IBD patients was lower than in healthy controls [80].

Correspondingly, the MOP/NOP mixed agonist BU08070 had been evaluated in mouse models mimicking IBS symptoms such as diarrhea and abdominal pain. This compound showed an anti-transit and anti-diarrheal potency in the mouse model of castor oil-induced diarrhea [74]. The results revealed that BU08070 inhibits smooth muscle contractility in vitro and has a strong inhibitory effect on GI motility [74]. The antinociceptive effect of BU08070 was blocked by the MOP antagonist β-FNA but not the NOP antagonist J-113397. In the mouse model of abdominal pain induced by the i.c. injection of MO, BU08070 (0.1 mg/kg, i.p.) significantly reduced pain-induced behaviors such as the licking of the abdomen, stretching the abdomen, the squashing of the lower abdomen against the floor and abdominal retraction [74].

NOP receptor modulation was also determined during intestinal inflammation. The effect of BU08070 (1 mg/kg i.p.) was tested in the mouse model of TNBS-induced colitis, and BU08070 was found to significantly reduce the severity of colitis compared to controls. The anti-inflammatory effect of BU08070 was reversed by selective NOP and MOP receptor antagonists [83]. Accordingly, nociceptin and the NOP antagonist UFP-101 were used in the rat model of colitis induced by TNBS [50]. In studies of antagonism, TNBS-treated rats were injected i.p. with saline, UFP-101(1–3–10 nM/kg), N/OFQ (0.02–0.2–2–20 nM/kg) and UFP-101 + N/OFQ or saline twice a day for three consecutive days 24 h after TNBS administration. Additionally, myeloperoxidase (MPO) activity and IL-1β and IL-10 levels were measured, and it was concluded that repeated treatment with nociceptin significantly decreased IL-1β levels compared to TNBS-treated animals injected with saline, but neither nociceptin nor UFP-101 modified the pro-inflammatory cytokine levels in inflamed rats. Repeated i.p. injections of nociceptin, at doses of 0.02 and 0.2 nM/kg, significantly reduced MPO activity. On the other hand, nociceptin administered at a dose of 20 nM/kg induced a significant increase in colonic MPO activity compared with TNBS-treated rats with saline. The conclusion of the study was that peripheral low doses of nociceptin did indeed improved colitis but at higher doses it worsened inflammation [84]. The mechanism of this phenomenon is yet to be evaluated.

## 6. Conclusions

Based on the presented literature, there is a high likelihood that NOP receptors communicate with the ECS. The activation of NOP receptors as well as CB1, CB2 and non-classical cannabinoid receptors leads to Gi/o protein modulation through the inhibition of adenylate cyclase, modulation of ion flow through K^+^ and Ca^2+^ channels and concomitant inhibition of neurotransmitter release. Their molecular pathway clearly overlaps, and they share at least two kinases, with MAPK/ERK, p38, JNK, NF-κB and β-arrestin mediating their action. NF-κB-inducing kinase plays a key role in the regulation of inflammation, and it was established that NOP receptor antagonists decrease levels of proinflammatory cytokines and peripheral low doses of nociception soothe inflammation. The disturbance of acetylcholine and neuropeptide transmission in the enteric nervous system contributes to diarrhoea and constipation in IBD, hence NOP receptor activation leads to GI contractions and hypermotility inhibition. Moreover, ECS and NOP receptors influence high and low voltage calcium channels, which are crucial in pain processing. Based on the previously mentioned in-vitro and in vivo studies, it can be stated that such receptors play a key role in pain modulation and, considering the content of this review, especially visceral pain management. Therefore, fully exploring the interactions between CB and NOP receptors in colitis will create the basis for the novel treatment of IBD and other intestinal diseases.

## Figures and Tables

**Figure 1 jcm-11-06675-f001:**
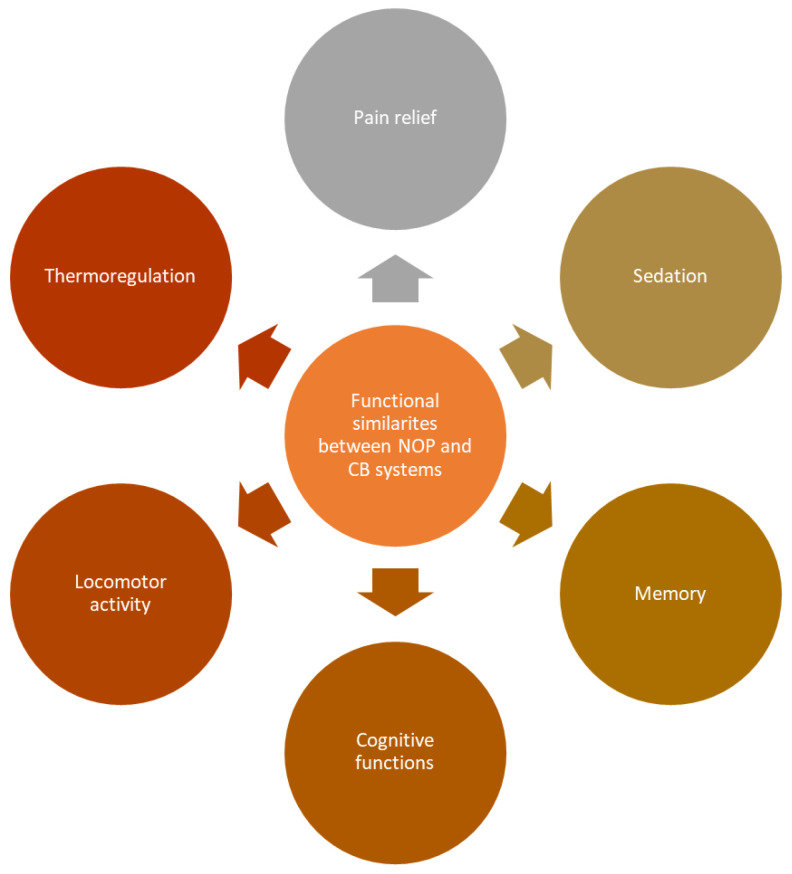
Functional similarities between NOP and CB systems.

**Figure 2 jcm-11-06675-f002:**
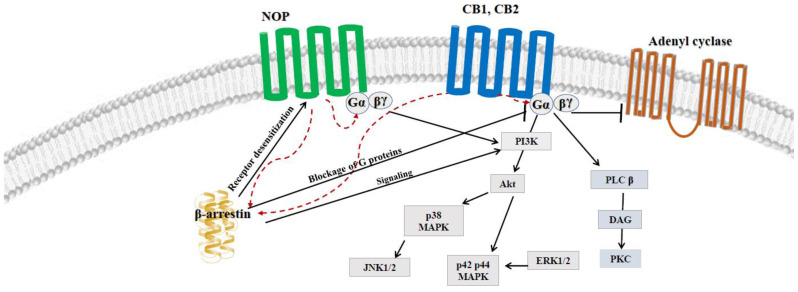
Shared elements of molecular pathways between NOP and CB1 and CB2 receptors.

## Data Availability

Not applicable.

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
