# Peer review of "A Crosstalk between the Cannabinoid Receptors and Nociceptin Receptors in Colitis—Clinical Implications"

_jcm, 2022, doi:10.3390/jcm11226675_

Round 1

Reviewer 1 Report

Review for the manuscript A cross-talk between the cannabinoid receptors and noci- 2 ceptin receptors in colitis – clinical implications.

            Dear authors, thank you for the opportunity to review this interesting manuscript. I have many suggestions for you.

ABSTRACT

In lines 10-11 we find, “Inflammatory bowel diseases refers to a group of the gastrointestinal disorders with complex pathogenesis characterized by chronic intestinal inflammation and abdominal pain”. I suggest modifying for “Inflammatory Bowel Diseases refer to a group of gastrointestinal disorders with complex pathogenesis characterized by chronic intestinal inflammation and a variety of symptoms that may include abdominal pain.”

            Please, inform in this section that it is a review.

            I suggest re-building the abstract for a more informative one.

KEYWORDS

In this part we find “Nociceptin Opioid Receptor, NOP, Encocannabinoid, ECS, colitis, IBD, inflammation”. Please, change:

·       Encocannabinoid for Endocannabinoid,

·       IBD for Inflammatory Bowel Diseases.

·       Do not use NOP and ECS, please, define the acronyms.

INTRODUCTION

            In lines 23-31: Please, include references to this paragraph.

            There is only one citation in this section, which is insufficient to support what the authors wrote. In addition, the one used is from 2014 (authors can find many references on the subject on PUBMED and much more current than this one). Please include references to all information explained in the Introduction.

            This section contains sentences without references (an example is in lines 56-58). Please, include.

            Lines 81-84 we can read

“Research based on surveys, collected among Spanish patients have reported 40-50% rates of CAM usage among IBD patients [14]. Ten per cent of them admitted to take mainly cannabis derivatives. Interestingly, patients diagnosed with UC tended to a higher rate of CAM usage [14] than CD patients”

I suggest modifying for: “Research based on surveys, collected among Spanish patients have reported 40-50% rates of CAM usage among IBD patients [14]. Ten percent of them admitted to taking mainly cannabis derivatives. Interestingly, patients diagnosed with UC tended to a higher rate of CAM usage than CD patients [14].”

In lines 102-104, the authors mention that “Salaga et al…” and at the end of the sentence, they cite [21,22]. However, when we go to the references section, we find:

[20] Sałaga M, Polepally PR, Zakrzewski PK, Cygankiewicz A, Sobczak M, Kordek R, et al. Novel orally available salvinorin A analog PR-38 protects against experimental colitis and reduces abdominal pain in mice by interaction with opioid and cannabinoid receptors. Biochemical Pharmacology. 2014 Dec 15;92(4):618–26. 480 21.

[21] Fichna J, Sobczak M, Mokrowiecka A, Cygankiewicz AI, Zakrzewski PK, Cenac N, et al. Activation of the endogenous nociceptin system by selective nociceptin receptor agonist SCH 221510 produces antitransit and antinociceptive effect: a novel strategy for treatment of diarrhea-predominant IBS Address for Correspondence Neurogastroenterology & Motility. Neurogastro- 483 enterol Motil. 2014;26:1539–50.

Please, verify the correct citations for this sentence.

In lines 155-158 we can find “Derkinderen et al. showed that lysophosphatidic acid in- 155 creased phosphorylation of p38 MAPK and c-Jun N-terminal kinase in rat and mouse hip- pocampal slices and the pre-treatment with selective antagonist of CB1, SR 141716 at the concentration of 100 µM per slice abolished these effects [38].” However, reference 38 in the references section is Glass et al 1997. Derkinderen et al is reference 40.

In line 182-186 we find “Donica et al. found that nociceptin modulates NF-κB activity on SH-SY5Y cells, as the nuclear accumulation and the DNA binding of NFκB is both protein kinase C-dependent and NOP receptors- specific. Moreover, NF-κB inhibitors interfered with the effects of lipopolysaccharide-induced expression of nociceptin, and NFκB subunits p50 and p65 were shown to form a heterodimer which is 186 implicated in inflammatory pathways [44,45]”. There are two citations for Donica et al in the references section, but they are not 44, 45. Besides that, if you want to cite two studies by the same author, you should cite “Donica et al [xxx] and Donica et al [yyy].

Again in lines 269-271: “In a study by Cichewicz et al. regarding correlation between formation of morphine 269 tolerance and levels of CB1 receptor protein, no apparent changes in amount of CB1 were 270 described in midbrain regions, and there was an up-regulation of CB1 protein in the spinal 271 cord [67].” Cichewicz et al. do not correspond to reference 67.

There are many problems with the citations along with the text. I suggest the authors carefully revise all the citations along with the text and compare them with the references section. Maybe you could use a citation manager to avoid these problems.

In Lines 291-293 we see “Interestingly, the hypothermic 291 effect of nociceptin was blocked by JTC-801, but not by selective SR 141716A “. There is a reference missing here.

Along with the text, use “in vivo and in vitro” instead of “in vivo and in vitro”

See in Table 1: there is a “.” After references [36,37]. This table is not mentioned in the text. There is not a legend for it. I do not see a reason for keeping this table in the text.

FINAL COMMENTS:

I suggest authors update references throughout the text. There is only one quote from 2022, one from 2021, three from 2020, and three from 2019. All the others are older than that. I understand that there are excellent studies that are not new. However, it is also necessary, in a review, to show what is new.

Review the citations along with the text. If there are many authors numbered differently from the References section, may many other citations are not correct after each sentence.

Please include the search strategy and mesh terms used to build this review. What were databases used? Was there a period restriction? Were there exclusion and inclusion criteria to build the review?

Author Response

We would like to express our gratitude for all the valuable comments provided. All comments have been addressed accordingly.

ABSTRACT

  1. In lines 10-11 we find, “Inflammatory bowel diseases refers to a group of the gastrointestinal disorders with complex pathogenesis characterized by chronic intestinal inflammation and abdominal pain”. I suggest modifying for “Inflammatory Bowel Diseases refer to a group of gastrointestinal disorders with complex pathogenesis characterized by chronic intestinal inflammation and a variety of symptoms that may include abdominal pain.”

It has been corrected according to reviewer’s suggestions

  1. Please, inform in this section that it is a review.

It has been corrected according to reviewer’s suggestions

  1. I suggest re-building the abstract for a more informative one.

 We re-phrased abstract and added new information to make it more informative.

KEYWORDS

In this part we find “Nociceptin Opioid Receptor, NOP, Encocannabinoid, ECS, colitis, IBD, inflammation”. Please, change:

  • Encocannabinoid for Endocannabinoid,
  • IBD for Inflammatory Bowel Diseases.
  • Do not use NOP and ECS, please, define the acronyms.

It has been corrected according to reviewer’s suggestions.

INTRODUCTION

  1. In lines 23-31: Please, include references to this paragraph.

It has been included in the text according to reviewer’s suggestions.

  1. There is only one citation in this section, which is insufficient to support what the authors wrote. In addition, the one used is from 2014 (authors can find many references on the subject on PUBMED and much more current than this one). Please include references to all information explained in the Introduction.

The missing references were added. This reference from 2014 is concerning variability of factors that contribute to IBD development and until now there is an agreement that it is multifactorial disease of unknown etiology and current research is carried out to understand the ethiology of the colitis.

  1. This section contains sentences without references (an example is in lines 56-58). Please, include.

It has been corrected according to reviewer’s suggestions.

  1. Lines 81-84 we can read

“Research based on surveys, collected among Spanish patients have reported 40-50% rates of CAM usage among IBD patients [14]. Ten per cent of them admitted to take mainly cannabis derivatives. Interestingly, patients diagnosed with UC tended to a higher rate of CAM usage [14] than CD patients”

I suggest modifying for: “Research based on surveys, collected among Spanish patients have reported 40-50% rates of CAM usage among IBD patients [14]. Ten percent of them admitted to taking mainly cannabis derivatives. Interestingly, patients diagnosed with UC tended to a higher rate of CAM usage than CD patients [14].”

It has been modified according to reviewer’s suggestions.

  1. In lines 102-104, the authors mention that “Salaga et al…” and at the end of the sentence, they cite [21,22]. However, when we go to the references section, we find:

[20] Sałaga M, Polepally PR, Zakrzewski PK, Cygankiewicz A, Sobczak M, Kordek R, et al. Novel orally available salvinorin A analog PR-38 protects against experimental colitis and reduces abdominal pain in mice by interaction with opioid and cannabinoid receptors. Biochemical Pharmacology. 2014 Dec 15;92(4):618–26. 480 21.

[21] Fichna J, Sobczak M, Mokrowiecka A, Cygankiewicz AI, Zakrzewski PK, Cenac N, et al. Activation of the endogenous nociceptin system by selective nociceptin receptor agonist SCH 221510 produces antitransit and antinociceptive effect: a novel strategy for treatment of diarrhea-predominant IBS Address for Correspondence Neurogastroenterology & Motility. Neurogastro- 483 enterol Motil. 2014;26:1539–50.

Please, verify the correct citations for this sentence.

It has been revised and corrected according to reviewer’s suggestions.

  1. In lines 155-158 we can find “Derkinderen et al. showed that lysophosphatidic acid in- 155 creased phosphorylation of p38 MAPK and c-Jun N-terminal kinase in rat and mouse hip- pocampal slices and the pre-treatment with selective antagonist of CB1, SR 141716 at the concentration of 100 µM per slice abolished these effects [38].” However, reference 38 in the references section is Glass et al 1997. Derkinderen et al is reference 40.

In line 182-186 we find “Donica et al. found that nociceptin modulates NF-κB activity on SH-SY5Y cells, as the nuclear accumulation and the DNA binding of NFκB is both protein kinase C-dependent and NOP receptors- specific. Moreover, NF-κB inhibitors interfered with the effects of lipopolysaccharide-induced expression of nociceptin, and NFκB subunits p50 and p65 were shown to form a heterodimer which is 186 implicated in inflammatory pathways [44,45]”. There are two citations for Donica et al in the references section, but they are not 44, 45. Besides that, if you want to cite two studies by the same author, you should cite “Donica et al [xxx] and Donica et al [yyy].

Again in lines 269-271: “In a study by Cichewicz et al. regarding correlation between formation of morphine 269 tolerance and levels of CB1 receptor protein, no apparent changes in amount of CB1 were 270 described in midbrain regions, and there was an up-regulation of CB1 protein in the spinal 271 cord [67].” Cichewicz et al. do not correspond to reference 67.

There are many problems with the citations along with the text. I suggest the authors carefully revise all the citations along with the text and compare them with the references section. Maybe you could use a citation manager to avoid these problems.

The references, their style and reference list have been carefully revised and corrected as suggested by reviewer. We used Mendeley to perform our reference list, however there was unexpected problem during removing field codes, which we did not see during uploading files to the MDPI system.

  1. In Lines 291-293 we see “Interestingly, the hypothermic 291 effect of nociceptin was blocked by JTC-801, but not by selective SR 141716A “. There is a reference missing here.

The missing reference was been added according to reviewer’s suggestions.

  1. Along with the text, use “in vivoand in vitro” instead of “in vivo and in vitro”

It has been corrected for italic fonts according to reviewer’s suggestions.

  1. See in Table 1: there is a “.” After references [36,37]. This table is not mentioned in the text. There is not a legend for it. I do not see a reason for keeping this table in the text.

This table has been removed from our manuscript as suggested by Reviewer.

FINAL COMMENTS:

I suggest authors update references throughout the text. There is only one quote from 2022, one from 2021, three from 2020, and three from 2019. All the others are older than that. I understand that there are excellent studies that are not new. However, it is also necessary, in a review, to show what is new.

Review the citations along with the text. If there are many authors numbered differently from the References section, may many other citations are not correct after each sentence.

Please include the search strategy and mesh terms used to build this review. What were databases used? Was there a period restriction? Were there exclusion and inclusion criteria to build the review?

Literature search was performed with a use of the following keywords: cannabinoids, nociceptin, cross-talk and also in with a combination with of the essential terms for the article topic, e.g.: nociceptin and inflammation, cannabinoids and colitis. All generated results were filtered by their relevance, sorted by date, and positions published in a period of 2010-2022 are mainly cited, nonetheless crucial to the topic understanding publications were also referenced.

The review citations were reviewed and corrected.

Reviewer 2 Report

Abstract:

-       After the first inflammatory bowel disease is mentioned, authors should add an abbreviation.

-       Revise the sentence starting with “Endocannabinoid receptors and ligands are widely distributed.”

-       What are the functional similarities? Authors should add a sentence that will help to understand functional similarities.

Keywords:

-       The abbreviation should be removed from keywords.

Introduction:

-       The first sentence needs a citation.

-       Add citation to following sentence “Moreover, co-administration of cannabinoids and opioids was proven to 28 produce synergistic effects in pain management.

-       Use different words instead plausibility in lines 30 and 40.

-       Revise the following sentence “The ECS consists of classical receptors, CB1 and CB2, their endogenous ligands endocannabinoids and various compounds and proteins involved in modulation, synthesis 49 and degradation of the latter”

-       Use italic fonts when mentioning plants

-       Revise the word “analgetic”

-       The following sentence need citation “However, 76 data on the efficacy of the treatment using cannabis are deficient and inconsistent, and 77 therapy itself remains controversial.”

-       Use different words for these days on line 78

-       Use a uniform citation format throughout the manuscript, which is already stated on the journal’s website.

-       Revise the following sentence “NOP receptors became a point of interest of various 113 research groups, and it was claimed to play a role in the food intake by elevation of food 114 consumption, when administered centrally [24], prolonged stress and post-traumatic 115 stress disorder [25], learning [26] and addictions, including alcohol abuse [27].”

-       The following sentence needs citation “The analysis of 3.0A resolution X-ray structure revealed the key distinction 128 between NOP receptors and opioid receptors.”

-       Authors should revise and double-check the manuscript for typos.

-       Authors should provide illustrations to improve their manuscript.

Author Response

We would like to express our gratitude for all the valuable comments provided. All comments have been addressed accordingly.

Abstract:

-       After the first inflammatory bowel disease is mentioned, authors should add an abbreviation.

-       Revise the sentence starting with “Endocannabinoid receptors and ligands are widely distributed.”

-       What are the functional similarities? Authors should add a sentence that will help to understand functional similarities.

Keywords:

-       The abbreviation should be removed from keywords

The Abstract section was modified according to the Reviewer’s suggestion. The abbrevations have been removed from the keywords.

Introduction:

-       The first sentence needs a citation.

The citations have been added to the section of introduction.

-       Add citation to following sentence “Moreover, co-administration of cannabinoids and opioids was proven to 28 produce synergistic effects in pain management.

The missing reference has been added in the indicated sentence.

-       Use different words instead plausibility in lines 30 and 40.

It has been changed as suggested by reviewer.

-       Revise the following sentence “The ECS consists of classical receptors, CB1 and CB2, their endogenous ligands endocannabinoids and various compounds and proteins involved in modulation, synthesis 49 and degradation of the latter”

This statement has been corrected and in current form we hope it is acceptable.

-       Use italic fonts when mentioning plants

The names which required italic fonts have been corrected.

-       Revise the word “analgetic”

It has been corrected to analgesic.

-       The following sentence need citation “However, 76 data on the efficacy of the treatment using cannabis are deficient and inconsistent, and 77 therapy itself remains controversial.”

The citation has been added to the manuscript.

-       Use different words for these days on line 78

It has been changed according to reviewer’s suggestions.

-       Use a uniform citation format throughout the manuscript, which is already stated on the journal’s website.

-       Revise the following sentence “NOP receptors became a point of interest of various 113 research groups, and it was claimed to play a role in the food intake by elevation of food 114 consumption, when administered centrally [24], prolonged stress and post-traumatic 115 stress disorder [25], learning [26] and addictions, including alcohol abuse [27].”

The references and reference list have been carefully corrected according to reviewer’s suggestions.

-       The following sentence needs citation “The analysis of 3.0A resolution X-ray structure revealed the key distinction 128 between NOP receptors and opioid receptors.”

This citation has been added to the manuscript.

-       Authors should revise and double-check the manuscript for typos.

This manuscript has been read by native speaker and typos have been corrected according to reviewer’s suggestions.

-       Authors should provide illustrations to improve their manuscript.

We have added new figure, which helps to show the molecular functions shared by nociceptin and cannabinoid receptors under physiological and pathophysiological conditions.

Round 2

Reviewer 1 Report

Dear authors,
Thank you for performing the corrections.

With best regrds

Reviewer 2 Report

The authors made proper revisions.